

# Multiresolution wavelet analysis applied to GRACE range rate residuals

Saniya Behzadpour[1,2], Torsten Mayer-Gürr[1], Jakob Flury[2], Beate Klinger[1], and Sujata Goswami[3]

[1]Graz University of Technology, Institute of Geodesy, Steyrergasse 30/III, 8010 Graz, Austria
[2]Leibniz University of Hanover, Institute of Geodesy, Schneiderberg 50, 30167 Hanover, Germany
[3]Leibniz University of Hanover, Max Planck Institute of Gravitational Physics, Callinstrasse 38, 30167 Hanover, Germany

**Correspondence:** Saniya Behzadpour (behzadpour@tugraz.at)

**Abstract.** For further improvements of gravity field models based on Gravity Recovery and Climate Experiment (GRACE) observations, it is necessary to identify the error sources within the recovery process. Observation residuals obtained during the gravity field recovery contain most of the measurement and modeling errors and thus can be considered as a realization of actual errors.

In this work, we investigate the ability of wavelets to help in identifying specific error sources in GRACE range rate residuals. The Multi-Resolution Analysis (MRA) using Discrete Wavelet Transform (DWT) is applied to decompose the residual signal into different scales with corresponding frequency bands. Temporal, spatial, and orbit-related features of each scale are then extracted for further investigations.

The wavelet analysis has proved to be a practical tool to find the main error contributors. Beside the previously known

sources such as K-Band Ranging (KBR) system noise and systematic attitude variations, this method clearly shows effects which the classic spectral analysis is hardly able or unable to represent. These effects include long-term signatures due to satellite eclipse crossings and dominant ocean tide errors.

*Copyright statement.* TEXT

# 1   Introduction

For more than 15 years, the Gravity Recovery and Climate Experiment (GRACE) satellite mission measured the time variation

of Earth's gravity field with a high temporal and spatial resolution (Tapley et al., 2004). The mission was a trailing formation of two satellites, GRACE-A and GRACE-B and provided the observation signals of inter-satellite ranging, GPS tracking, the satellites' attitude, and non-gravitational accelerations, which are required for the gravity field parameter estimation.

Based on these observations, various time-variable gravity models with monthly resolution are published by different analy-

sis centers (e.g., Bettadpur, 2012; Dahle et al., 2012; Mayer-Gürr et al., 2016). The accuracy level of such models has gradually increased in recent years; however it has not reached the GRACE baseline accuracy computed through pre-launch simulations

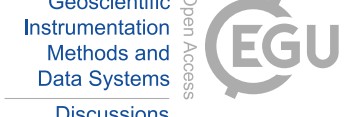



(Kim, 2000; Kim and Tapley, 2002). This results in an ongoing effort to understand the error content of GRACE observations, as well as any inaccuracies in the physical and stochastic models used for processing GRACE data.

In recent years, significant research efforts have been made to identify and parametrize the systematic errors such as uncertainties in star camera alignment (Bandikova and Flury, 2014; Harvey, 2016) and accelerometer calibration (Klinger and Mayer-Gürr, 2016). Along with the effects of geophysical aliasing and uncertainties in background models, these errors propagate through the numerical estimation of a large number of parameters. These parameters include gravity parameters in terms of spherical harmonics coefficients as well as orbit and sensor calibration parameters (Mayer-Gürr, 2013).

If the calibration parameters are correctly adjusted and the stochastic model fully describes the observation noise, it is expected that all of the mentioned errors are completely contained within the residuals. In reality, however, these errors might affect the gravity parameters due to imperfections in modeling. Therefore, residual analysis becomes a research topic as it is not only a way to study measurement and physical modeling errors, but also helps to evaluate and improve the gravity field solutions.

The studies in this field have been conducted mainly on the theoretical residuals, which are the difference between the actual GRACE ranging observation and simulated observation computed through force models. Ditmar et al. (2012) applied spectral analysis on theoretical residuals and showed that the major contributes to the noise budget at high frequencies are K-Band Ranging (KBR) sensor noise and inaccuracies in Earth's assumed static gravity field at higher degrees. It also has been shown that uncertainties in background models and errors in computed dynamic orbits contribute to low-frequency noise.

The main challenge in the spectral analysis of the residuals is that several noisy signals and disturbances are known to be superimposed at each frequency. Furthermore, the analysis is based on the assumption of the stationary behavior of these signals. However, in reality, most of these signals have non-stationary behavior, meaning that they have dynamic frequency components over time. Classical spectral analysis using Fourier transforms only represents the frequency content of such signals and does not provide any information about the time at which a signal at a specific frequency occurred or the duration for which it lasted (Keller, 2004). Consequently, in this framework, it is not possible to localize each component of the residuals in time for further statistical, spatial, or orbital analysis.

In an attempt to consider time variations in the sought-after signals, time-frequency methods can be applied to identify and localize the content of the non-stationary signals in the time and frequency domains simultaneously. The simplest method is the Short Time Fourier Transform (STFT), which is implemented by sliding a window throughout a signal and applying a Fourier transform to each windowed data segment. The squared magnitudes of the STFT coefficients form a spectrogram, representing the variation of the signal's spectrum over time (Fig. 1b). The shape and length of the window function determines the fixed time and frequency resolution of the STFT. Due to this uniform time resolution for all frequencies, the STFT is limited in capturing time information of rapid changes in a signal as well as spectral information in its lower frequency components.

To overcome STFT drawbacks, the wavelet analysis was introduced as a more effective technique for representation, decomposition, and reconstruction of non-stationary signals (Keller, 2004). In contrast to STFT, the wavelet transform provides a better trade-off between time and frequency resolution by using windows with shorter timespans at higher frequencies and windows with longer timespans at lower frequencies. The Multi-Resolution Analysis (MRA), introduced by Mallat (1989)





and Meyer (1993), is an efficient implementation of a wavelet transform for real signals. MRA can decompose a signal into multiscale components which can describe all time-variable structures in that signal.

The aim of this paper is to exploit the advantages of the wavelet transform to investigate the major contributors to GRACE range rate residuals and ideally detect non-stationary noise sources in sensors and background models which cannot be ob-
served with traditional spectral analysis. The results of this study will further improve gravity field modeling based on GRACE data. In addition, they will be beneficial for the preparation of GRACE Follow-On data processing infrastructure. To reach this goal, we decompose the residual signal into three groups of scale and compare the characteristics of each group with known or supposed sources.

In the upcoming section 2, we explain how the residual signal is obtained in the frame of computing the ITSG-Grace2016
model and review the performed data processing steps in order to introduce potential error sources. Section 3 discusses the methodology of the multi-resolution analysis and the wavelet transform. In section 4, results of the employed method on the residuals are described. Finally, section 5 presents the interpretation of results and a discussion.

## 2   Range rate residuals from ITSG-Grace2016 model

In this study, we use GRACE range rate residuals obtained in the course of computing the ITSG-Grace2016 (Mayer-Gürr et al.,
2016) gravity field model up to degree and order 60. Therefore, in order to introduce the residual signal, we briefly explain the processing chain of the model (Klinger et al., 2016).

In the ITSG-Grace2016 gravity field processing, high-precision kinematic orbits (Zehentner and Mayer-Gürr, 2013) with a sampling of 5 minutes and K-band intersatellite range rates with a sampling of 5 seconds serve as observations. Using the approach of variational equations, dynamic orbits are computed for each day (Ellmer and Mayer-Gürr, 2017), and normal
equations are set up with an arc length of 3 hours. The accumulated normal equations are then solved to estimate gravity parameters in terms of spherical harmonic coefficients, spanning from degree 2 up to degree 60, 90 and 120. The background models used during the dynamic orbit integration are listed in Table 1.

In the course of the adjustment process, non-gravity parameters are also co-estimated for each day. These parameters include the initial orbit states of both satellites, accelerometer scale factor matrices, accelerometer biases modeled by cubic splines with
6-hour nodes, and daily gravity field variations up to degree and order 40.

It is worth mentioning that unlike in the standard GRACE monthly solutions, in ITSG-Grace2016 the correlations between observations within a data block of 3 hours are taken into account. For each observation type, a stochastic model of the observation noise is built under the assumption of stationarity. This model is estimated once per month directly from the observation residuals.
The weights for the different frequency components of the observations are determined through the residuals' PSD. This PSD is iteratively computed directly from the residuals through Variance Component Estimation (VCE) (Koch, 1999). VCE is also used to estimate the relative weights for the combination of different data types, i.e. kinematic orbits and range rate observations. This modeling approach seems to appropriately separate the complex colored noise in the observations from the





gravity signal; therefore we expect the residuals to contain most of the imperfections caused by the instruments and background models.

## 3  Multi-Resolution Analysis (MRA)

The wavelet transform $Wf(u,s)$ of a signal $f(t) \in L^2(\mathbb{R})$,

$$Wf(u,s) = \langle f, \psi_{u,s} \rangle = \int\limits_{-\infty}^{\infty} f(t) \frac{1}{\sqrt{s}} \bar{\psi}\left(\frac{t-u}{s}\right) \mathrm{d}t, \tag{1}$$

is the decomposition of that signal over a set of scaled and translated versions of a finite energy and normalized function, the mother wavelet $\bar{\psi}$.

For the wavelet transform $Wf(u,s)$, the translation parameter $u$ determines the location of the wavelet in the time domain, while the scale parameter $s$ is related to the frequency location. These parameters are continuous real values, therefore an infinite number of coefficients are needed to describe a signal in this framework. In a practical implementation, it is convenient to discretize these parameters, as the real signals are band limited. The usual choice is to follow a $J$-scale dyadic discretization based on powers of two. This transform is then called a Discrete Wavelet Transform (DWT). For a signal with sampling frequency of $F_s$, the resulting coefficients $d(j,n)$ can be interpreted as detail subsignal at the scale $2^j$ ($1 \le j \le J$), corresponding to the frequency interval $[F_s/2^{j+1}, F_s/2^j]$ :

$$d(j,n) = \sum_t f(t)\psi_{j,n}(t), \quad \text{with } \psi(j,n) = 2^{-j/2}\bar{\psi}\left(\frac{t-n2^j}{2^j}\right); \quad j,n \in \mathbb{Z}. \tag{2}$$

The approximation of the signal at the scale $J$, which corresponds to the frequency interval $[0, F_s/2^{J+1}]$ is also given by:

$$a(J,n) = \sum_t f(t)\phi_{J,n}(t), \tag{3}$$

where $\phi(J,n)$ is the scaling function, associated with the wavelet function $\psi(j,n)$ .

The original signal can be reconstructed by adding all layers of details up to decomposition scale $J$ as well as the approximation subsignal:

$$f(t) = \sum_n a(J,n)\phi_{J,n}(t) + \sum_{j \le J}\sum_n d(j,n)\psi_{j,n}(t). \tag{4}$$

Mallat (1989) showed that for a discrete signal $f[n]$, any DWT on the orthonormal basis of $L^2(\mathbb{R})$ could be characterized by a particular class of digital filters, the conjugate mirror filters. He introduced a fast discrete wavelet transform by implementing a pair of conjugate mirror filters, corresponding to a specific mother wavelet:

$$h[n] = \left\langle \frac{1}{\sqrt{2}}\bar{\phi}\left(\frac{t}{2}\right), \bar{\phi}(t-n) \right\rangle, \tag{5}$$



$$g[n] = \left\langle \frac{1}{\sqrt{2}} \bar{\psi} \left( \frac{t}{2} \right), \bar{\phi}(t-n) \right\rangle. \tag{6}$$

Mathematically, the convolution of the filter response with the discrete signal is expressed as follows:

$$a[p] = \sum_{n=-\infty}^{\infty} h[n-2p]f[n] = f \star \bar{h}[2p], \tag{7}$$

$$d[p] = \sum_{n=-\infty}^{\infty} g[n-2p]f[n] = f \star \bar{g}[2p]. \tag{8}$$

The scaling function, defined by the filter coefficients $h[n]$, provides approximation coefficients $a$, which are also referred to as low-pass output. The wavelet function, defined by the filter coefficients $g[n]$, provides the detail coefficients $d$, or alternatively the high-pass output. This decomposition step is followed by a factor two down-sampling of the output signals. According to Vetterli and Herley (1992), down-sampling cancels the aliasing between the resulting coefficients. This is a

10 necessary condition for recovery of the original signal with an inverse DWT.

A fast inverse DWT reconstructs the initial signal $f[n]$ by up-sampling and filtering. The up-sampling operation is done by inserting zeroes between every other coefficients in the output signals $a[n]$ and $d[n]$. The zero-padded coefficients $\hat{a}$ and $\hat{d}$ are then filtered by the corresponding inverse filters $\tilde{h}[n]$ and $\tilde{g}[n]$:

$$f[n] = \hat{a} \star \tilde{h}[n] + \hat{d} \star \tilde{g}[n]. \tag{9}$$

As described before, the DWT decomposes the original signal into an approximation subsignal and a detail subsignal. The MRA algorithm suggested by Mallat (1989) and Meyer (1993) calls for this decomposition to be repeated on the approximation subsignal, again yielding a detail subsignal and an approximation subsignal. The selection of the decomposition level depends on the initial size of the original signal, and the desired spectral and temporal resolution. Finally, the original signal can be represented by the approximation coefficients of the last decomposition level and the accumulated detail coefficients of all

decomposition levels. Figure 2 shows a 3-level decomposition MRA algorithm.

We applied MRA using a discrete Daubechies wavelet transform with 20 vanishing moments (Daubechies, 1992) to decompose a monthly time series of residuals into 8 different scales. The choice of the Daubechies wavelet is due to their usual application in signal detection and classification. The selection of a high vanishing moment is due to a high smoothness property of the resulting mother wavelet, leading to a better frequency localization in mHz-frequency band. Figure 3 shows scaling

and wavelet functions for Daubechies-20 together with its corresponding decomposition and reconstruction conjugate mirror filters.

As shown in Fig. 4, we merged detail coefficients into three major groups, defined approximately through three frequency subbands (Fig. 5):





(a) Short timescale details, containing the details at level 1 to 3, corresponding to the frequency band above 12.5 mHz;

(b) Medium timescale details, containing the details at level 4 to 5, corresponding to the frequency range from 3.125 mHz up to 12.5 mHz;

(c) Long timescale details, containing the details at level 6 to 8, corresponding to the frequency range from 0.391 mHz up to 3.125 mHz;

Each group is then reconstructed into a time series of residuals using Eq.(9). Afterward, the time series are analyzed in three different domains. We have chosen the domains in such a way that they highlight specific characteristics of the error sources contained within the residual time series. They are:

**Spectral/temporal domain** As mentioned in the first section, a spectrogram shows the variation of a signal's energy as a
10 function of time and frequency. Another tool which can be used directly on the wavelet coefficients is the scalogram, in which the amplitude of the coefficients are plotted as a function of the scale and transition parameters. In our analyses, we used spectrograms because the interpretation of a signal in terms of frequency is more accessible than in terms of scale (Fig. 6).

**Spatial domain** Plotting each time series with respect to the satellite ground-track is useful to identify any features of geo-
15 physical origin in the data (Fig. 7).

**Orbital domain** Plotting each time series as a function of satellite position and time reveals features related to the orbit geometry or instrument errors caused by orbital conditions. As the GRACE orbits are near circular, the position of each satellite can be specified without loss of accuracy by the argument of latitude, ranging from -180° to 180°. This domain represents the ascending equator pass of the satellite at 0°, the north pole at 90°, the descending equator pass at
20 180°/-180°, and then the south pole at -90° (Fig. 8).

These analyses are carried out on the entire residual time series from the ITSG-Grace2016 monthly solutions up to degree 60. Highlights of this analysis are presented in the next section.

## 4 Results

In order to prove if our applied method using the DWT is applicable to detect the error sources, we initially focused on the
25 investigation of known issues. For instance, it is known that the K-band system noise is dominant in the frequency range above 12.5 mHz. This frequency band corresponds to the short timescale details of the residuals. The power of the noise in this band increases linearly with frequency. This is a result of the way the range rate observations are derived from the range measurements by differentiation. Investigations by Ko et al. (2012) and Harvey et al. (2017) showed that the excessive high frequency signatures in this band are highly correlated with low SNR values of the K-band frequency observation by GRACE-
30 B. Figure 9 compares these SNR values with the wavelet short timescale components, revealing this strong correlation.



According to Bandikova et al. (2012), residuals due to errors in the satellite attitude determination and their effects on the computed antenna offset correction are also expected to be found in the mHz frequency band. Our time-frequency analysis shows a similarity between the residuals in medium timescale and the angular acceleration variations derived from star camera observations (Fig. 10). The spatial pattern of the residuals related to the attitude variations appears as horizontal bands (Fig. 7b), consistent with the results presented by Inácio et al. (2015).

These first investigations already show that our applied method is well-suited to identify error sources. However, compared to the spectral analysis, the advantage of the implemented method of DWT is a better separation of superimposed signals in frequencies lower than 12.5 mHz. This enabled the identification of (a) systematic errors caused by eclipse crossings of the satellites and (b) dominant ocean tide model errors, which are respectively explained in the following sections.

## 4.1 Satellite eclipse crossings

Analysis of the medium timescale details throughout the GRACE time span reveals long-term systematic signatures (Fig. 11a). Although the source of these errors is unknown, our investigation revealed a high correlation with the eclipse transit phases of GRACE-A and GRACE-B.

Each satellite passes through partial or full eclipse phases when it enters Earth's shadow. Occasionally the Moon also casts a shadow on the satellites. The eclipse factor is defined as the fraction of the Sun's light that reaches the satellite. It has a minimum value of zero if the satellite is in the umbra shadow of the occulting body and a maximum value of one if the satellite is in direct sunlight. For the detailed calculation, the reader is referred to Montenbruck and Gill (2000).

The difference between GRACE-B and GRACE-A eclipse factors indicates if the mission, i.e. one of the satellites, is in a transit mode. Difference values not equal to zero are interpreted as transit events, in which one of the satellites is passing through a partial eclipse phase. Figure 11b compares the medium timescale details of the residuals with the transit events for the complete GRACE timespan. Before 2011, the signatures are most obvious when the difference value is negative, meaning that GRACE-A is in the shadow and GRACE-B is in sunlight.

The GRACE formation mission started with GRACE-A as leading and GRACE-B as trailing satellite. After three years in orbit, the satellites had to exchange their position to limit the damage on the K-band horn caused by atomic oxygen. This swap maneuver happened at the end of 2005. Before this time, eclipse crossing signature occurs when the pair entered sunlight. After the orbit swap maneuver in December 2005, when GRACE-B became the leading satellite, the signatures are visible when the pair enters the shadow area.

However, after the year 2011, these rules cannot explain the eclipse crossing signatures in the residuals, as they appear in both entering and leaving shadow conditions with different intensities. The unstable thermal condition due to the disabled thermal controls might be a possible reason.

We compared the temperature measurements obtained from Level-1A High-Resolution Temperature data (HRT1A) for 2008-11 and 2011-10 with these signatures. It becomes obvious that there is a high correlation between the GRACE-B K-band antenna horn temperature variation and the disturbances during eclipse crossing events (Fig. 12). We suggest that the increasing temperature on the GRACE-B antenna horn may produce disturbances in the KBR measurements. This hypothesis can be



investigated in more details once the complete GRACE Level-1A data sets become publicly available. From a gravity field recovery point of view, these eclipse crossing signatures can be interpreted as a temporary unmodeled signal in the range rate measurements.

## 4.2 Ocean tide model

Errors in the background force models of temporal gravity field variations can be found in the long timescale details. Due to the spatial nature of these errors and the periodicity of satellite passes over their source regions, different model errors are superimposed at the same $n$ cycles-per-revolution frequencies. Therefore, frequency or time-frequency plots cannot differentiate the dominant source from other influences at this detail scale.

The two main potential error sources at this scale are (a) inaccuracies in the employed ocean tide model EOT11a (Savcenko and Bosch, 2012), and (b) inaccuracies in the employed non-tidal atmosphere and ocean mass variation model, AOD1B RL05 (Dobslaw et al., 2013). To better understand the contributions of the individual models, we swap in alternative models of the same forces and studied the resulting differences. This is best done in a closed-loop simulation, where other contributors to noise can be controlled. The following steps outline our employed simulation process:

1. Dynamic orbits are computed based on the background models mentioned in Table 1 with two exceptions. First, the FES2014 ocean tide model (Carrere et al., 2015) is substituted for the EOT11a model. Second, the AOD1B RL05 model and the van Dam-Ray atmospheric tide model (van Dam and Ray, 2010) were substituted with the AOD1B RL06 model. Compared to AOD1B RL05, the AOD1B RL06 model (Dobslaw et al., 2017) has undergone several improvements, amongst them a higher temporal resolution and the separation of non-tidal and tidal signals, including atmospheric tides with 12 selected frequencies. Therefore, there was no need to consider a dedicated atmospheric tide model in the simulation employing AOD1B RL06.

2. Error-free observations for position, velocity, non-gravitational accelerations, and the K-Band instrument are synthesized from these ideal orbits.

3. Realistic models of instrument noise are used to degrade synthesized observations. White Gaussian noise with a standard deviation of 3 cm is added to the simulated satellite's positions. Accelerometer observations are degraded by white noise with a standard deviation of 0.3 $\mathrm{nm\,s^{-2}}$ in along-track and radial directions and 3 $\mathrm{nm\,s^{-2}}$ in the cross-track direction. Star camera instrument noise is added as white Gaussian noise with a standard deviation of 0.05 mrad to the orientation quaternions. KBR instrument noise is computed by applying a differential filter to white Gaussian noise with a standard deviation of 0.25 $\mathrm{\mu m\,s^{-1}}$, which is then added to the simulated range rate observations.

4. The final step is to recover a monthly gravity field using the simulated degraded observations. To this end, the dynamic orbits are re-integrated using the artificially degraded accelerometer observations and the separate models under study, each in a dedicated scenario. The respective obtained residuals are then analyzed and compared.



### 4.2.1    Simulation Scenario 1: Propagated errors due to instrument noise

In the first scenario, the same background models as mentioned in the first step of the simulation process are used to compute the re-integrated dynamic orbits. Therefore the results only show the effects of instrument noise. As expected, the propagated noise is one order of magnitude smaller than the real residuals in frequency range from 0.391 mHz up to 3.125 mHz (Fig. 13b) and obviously cannot explain the errors in the long timescale details. Analyzing the solution in terms of RMS geoid heights per

degree with respect to the reference field GOCO05s, it can also be seen that the monthly solution based on instrument noise alone exhibits differences smaller than those of the GRACE baseline (Fig. 13a).

### 4.2.2    Simulation Scenario 2: Propagated errors due to AOD1B RL05

The second scenario studies the propagated errors due to inaccuracies of the non-tidal mass variation model. In order to recover a gravity field in this scenario, the AOD1B RL05 model and the van Dam-Ray atmospheric tide model (van Dam and Ray,

2010) were substituted for the AOD1B RL06 model. The simulated residual signal is then decomposed, and its long timescale components are compared to those obtained from real data. As shown in Fig. 13b, although the propagated errors have the same spectral behaviour at frequency range from 0.391 mHz to 3.125 mHz, their magnitude and spatial structure (Fig. 14a) cannot explain the real residuals.

### 4.2.3    Simulation Scenario 3: Propagated errors due to EOT11a

In the third scenario, we study the contribution of the ocean tide model. To recover a gravity field in this scenario, the EOT11a model ocean tide model is substituted for the FES2014 model. After decomposition of the simulated residual signal, its long timescale components are compared to the real data. These errors have the same magnitude and spatial pattern (Fig. 14b) as those in the real data (Fig. 14c). This leads to the conclusion that the ocean tide model is the dominant error source at the long timescale detail level.

These results showcase the capability of wavelet analysis in studying the signals due to geophysical processes in GRACE range-rate residuals. The implemented method efficiently finds structures in the signal which are not explicitly apparent in the PSD of the residuals. The wavelet analysis proves to be an efficient tool in decomposing the background model errors and finding the most prominent sources.

## 5    Discussion and conclusions

The results presented in this paper show the advantages of using a DWT in analyzing the range rate residuals from the ITSG-Grace2016 gravity field model. Several improvements in ITSG-Grace2016 resulted in a cumulative noise reduction of 20-40%, compared to its predecessor ITSG-Grace2014. The findings of this study will be useful in implementing further improvements in this series of gravity field models.

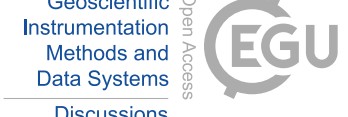


The proposed method presented results consistent with known systematic error sources in the residuals. We showed that the short timescale details of the residuals, equivalent to the frequency above 12.5 mHz, are dominated by KBR system noise. This is in agreement with the results presented by Ko et al. (2012) and Ditmar et al. (2012). The errors in the satellite attitude determination were also identified as a major contributor in the medium timescale details, equivalent to the frequency range from 3.125 mHz to 12.5 mHz. This finding is consistent with the results presented by Inácio et al. (2015) and Bandikova et al. (2012).

Besides the previously known instrument error sources, long-term signatures due to eclipse transits of the satellites were identified. They appear as a bias term in the K-Band range rate observations. As this is a clearly deterministic effect, its influence can be reduced by co-estimation of additional calibration parameters in the gravity field recovery process.

Analysis of the results from the implemented discrete wavelet transform brings new insights and a new understanding of the signals at the long timescale level. At this level, spectral analysis is unable to differentiate between the individual contributing sources, due to the non-stationary nature of the errors. Knowing that this scale level contains valuable information about the time-variable gravity field signal, we introduced non-tidal mass variation and ocean tide models as the potential dominant sources. Comparing simulation results with real data scenario, the EOT11a ocean tide errors are identified as the dominant error source within this scale. This means that using more accurate ocean tide model can lower the residuals in this frequency band.

It has been shown, that the wavelet-based MRA approach can properly represent the major error sources in GRACE processing data. These error sources have the largest impact on the accuracy of gravity field solutions derived from observations by GRACE. Even if the purpose of this study is to find the degrading factors in monthly gravity field models, which mainly are affecting the observations in mHz-frequency band, the investigation will be further continued by looking for physical interpretations for features at the lower frequencies of the residuals. This can be achieved by using a wavelet base with higher vanishing moments and thus higher decomposition level.

*Competing interests.* No competing interests are present.

*Acknowledgements.* We would like to thank the DFG Sonderforschungsbereich (SFB) 1128 Relativistic Geodesy and Gravimetry with Quantum Sensors (geo-Q) for financial support. Furthermore, we kindly thank Prof. Srinivas Bettadpur from the UTCSR for providing us the GRACE Level-1A test datasets.



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


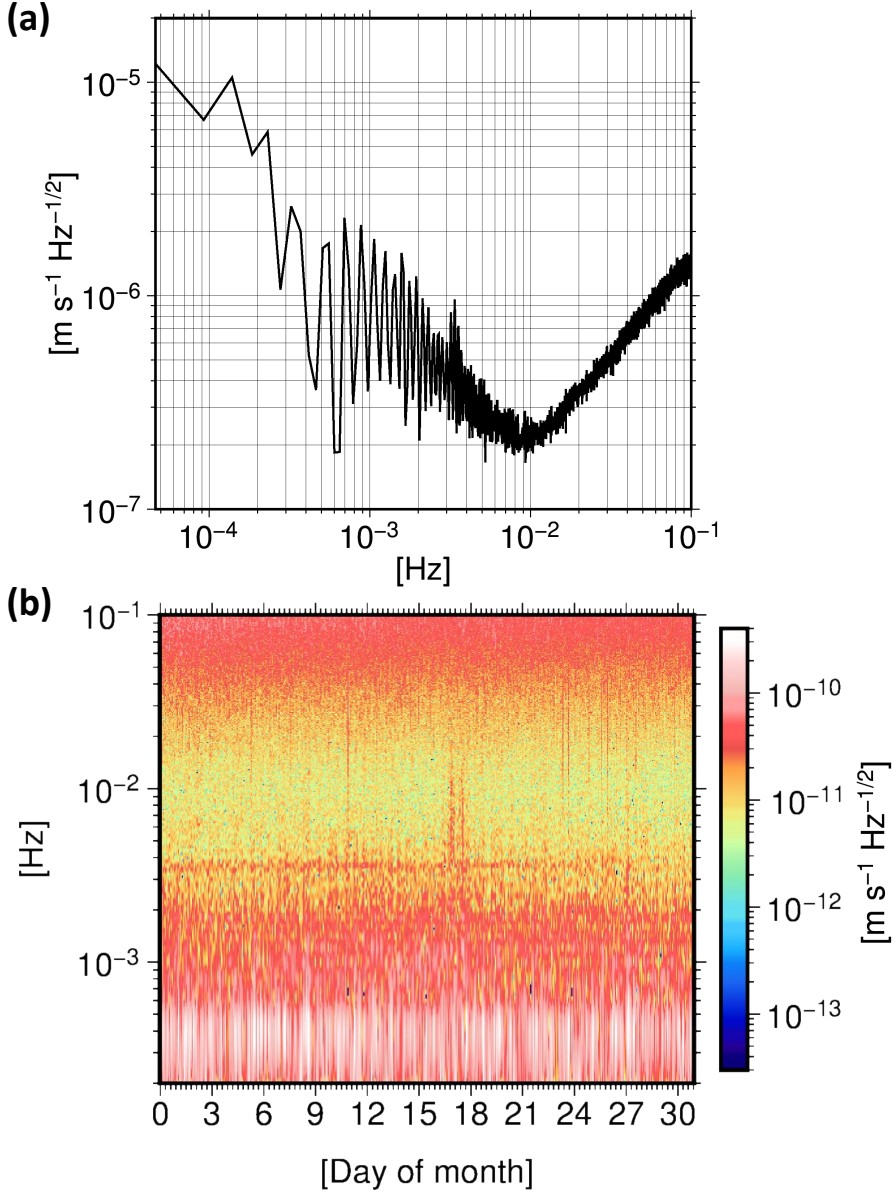

**Figure 1.** (a) PSD and (b) Spectrogram of the range rate residuals from December 2008. Time-frequency methods can be applied to the residual time series to localize the time variable frequency content.





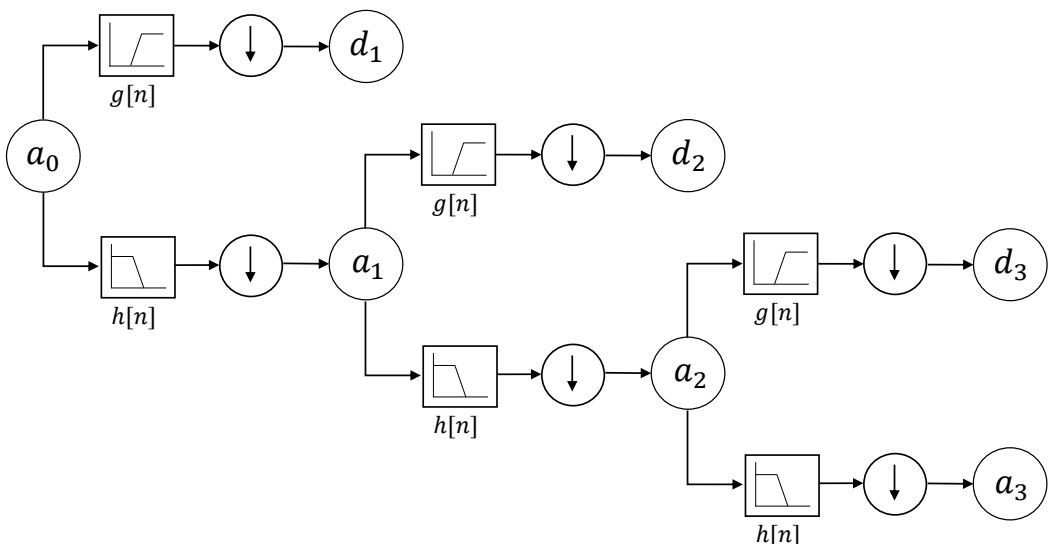

**Figure 2.** 3-level MRA decomposition tree, consisting of a high-pass filter $g[n]$ and a low-pass filter $h[n]$ followed by a downsampling operator at each level.




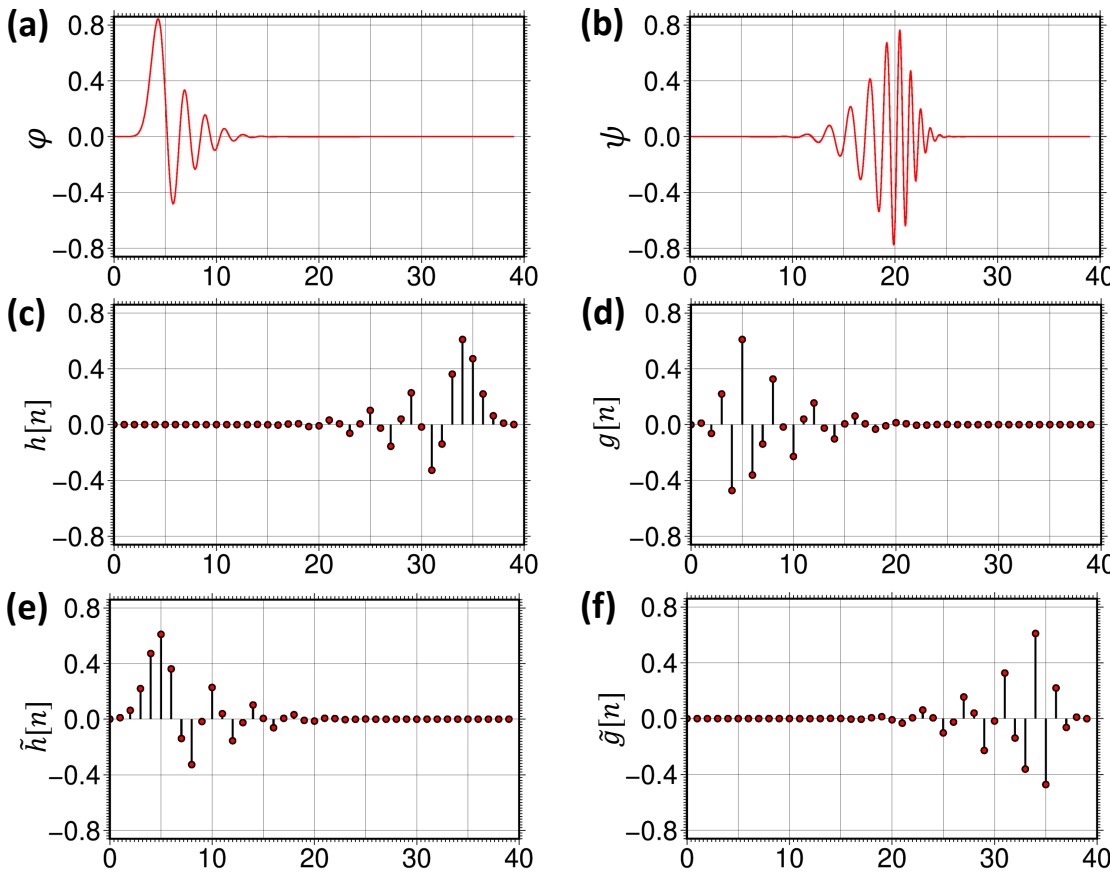

**Figure 3.** Daubechies-20 (a) scaling function, (b) wavelet function, (c) decomposition low pass filter, (d) decomposition high pass filter, (e) reconstruction low pass filter, and (f) reconstruction high pass filter.




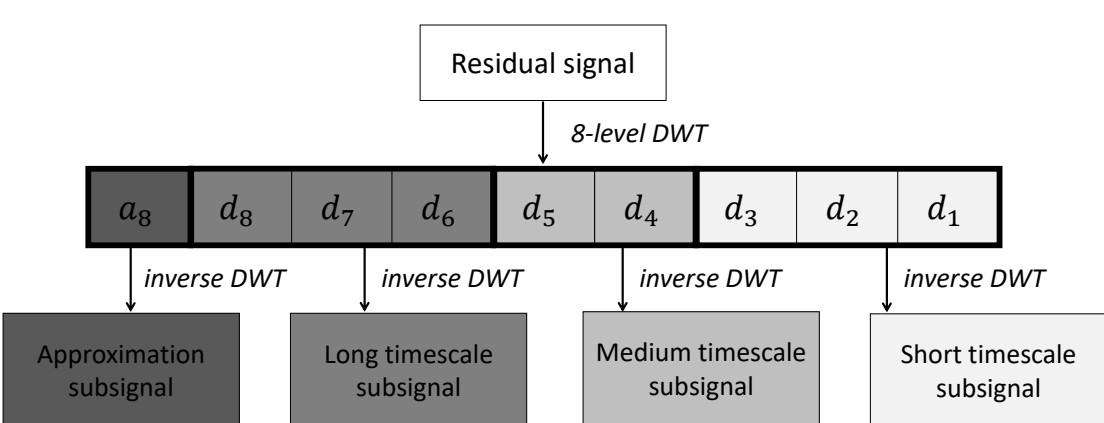

**Figure 4.** The proposed MRA scheme, implemented according to the characteristics of the residual signal.



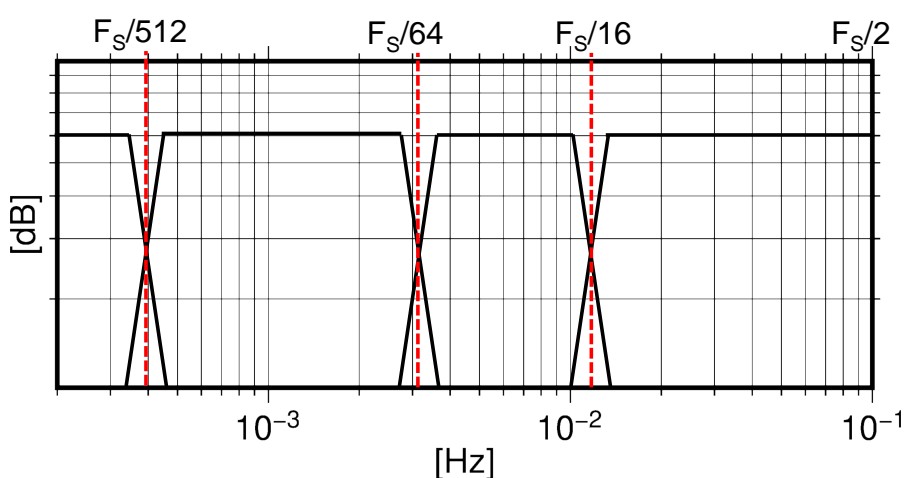

**Figure 5.** The proposed MRA bandwidth division of the residuals with frequency sampling $F_S$ of 0.2 Hz.





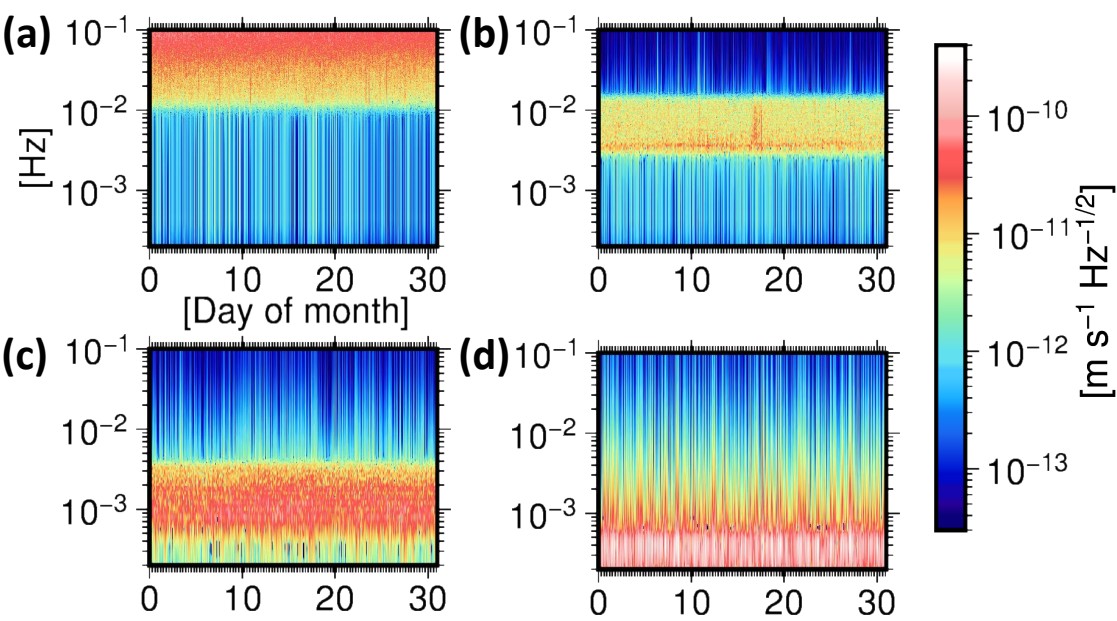

**Figure 6.** Time-frequency analysis of (a) short timescale, (b) medium timescale, (c) long timescale, and (d) approximation components of the residual signal. Spectrograms are computed with a window length of five hours for December 2008.





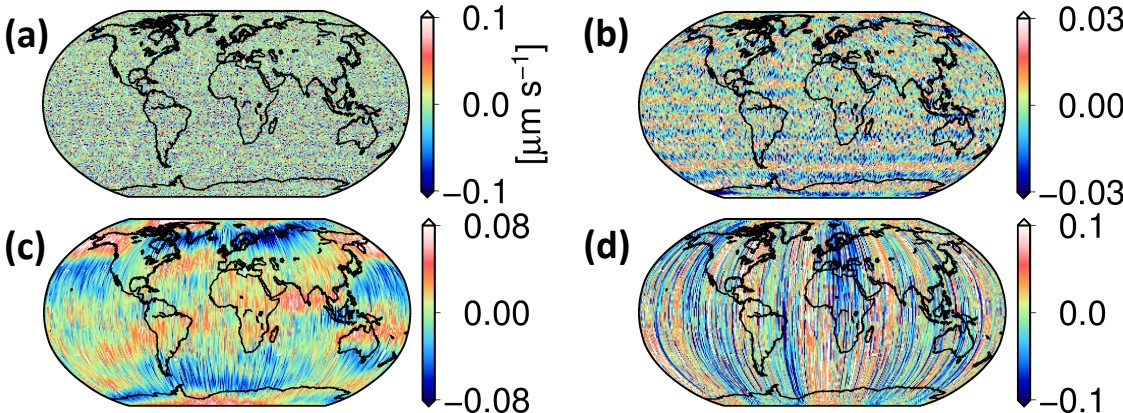

**Figure 7.** Spatial distribution of (a) short timescale, (b) medium timescale, (c) long timescale, and (d) approximation components of the residual signal. The values are plotted with respect to the GRACE-A ground-track for December 2008.

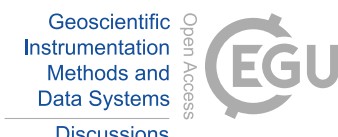



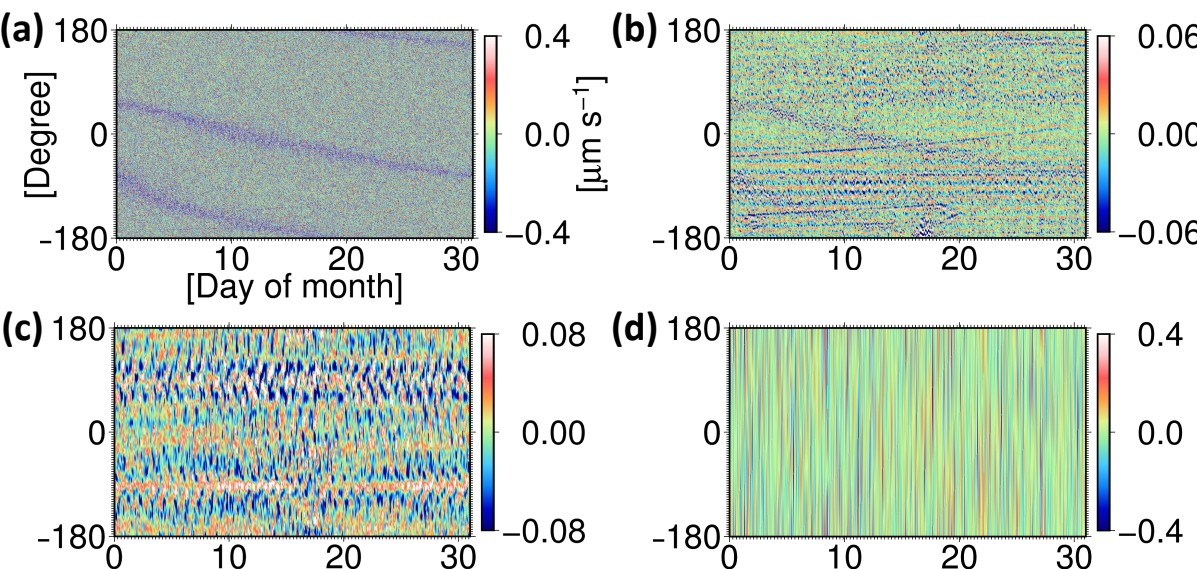

**Figure 8.** Orbital analysis of (a) short timescale, (b) medium timescale, (c) long timescale, and (d) approximation components of the residual signal. The values are plotted with respect to the GRACE-A argument of latitude for December 2008.



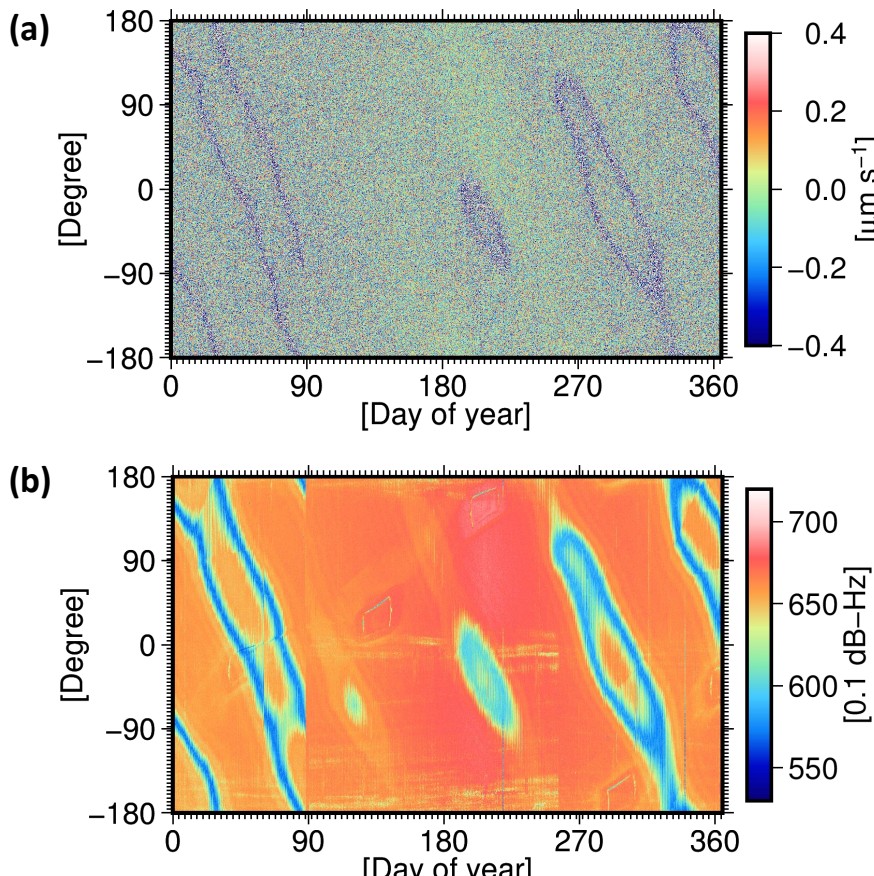

**Figure 9.** (a) Short timescale details of the residuals, (b) GRACE-B K-band SNR values. The values are plotted with respect to the GRACE-A argument of latitude for the time period 2009.



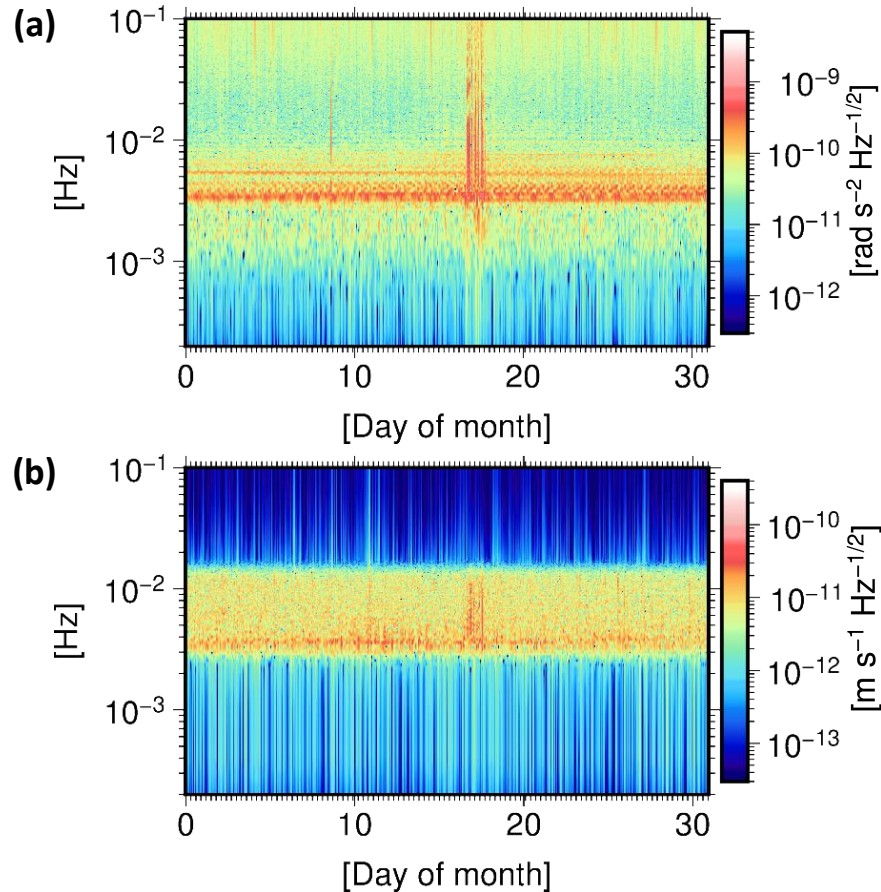

**Figure 10.** Spectrograms of (a) GRACE-A pitch angular acceleration variation and (b) medium timescale details of the residuals. The signal at 3.3mHz, which according to Bandikova et al. (2012) is induced by magnetic torquer attitude control, is clearly visible in the residuals for December 2008.





**Figure 11.** (a) Medium timescale details of the residuals and (b) the difference between GRACE-B and GRACE-A eclipse factors during the time period 2004-2010, plotted with respect to GRACE-A argument of latitude. The signatures are visible when the difference value is negative, i.e. GRACE-A is in the shadow and GRACE-B is in sunlight. (c) Medium timescale details of the residuals and (d) the difference between GRACE-B and GRACE-A eclipse factors during the time period 2011-2016, plotted with respect to GRACE-A argument of latitude. The signatures appear in both entering and leaving eclipse phase with different intensities. The gray areas indicate data gaps.


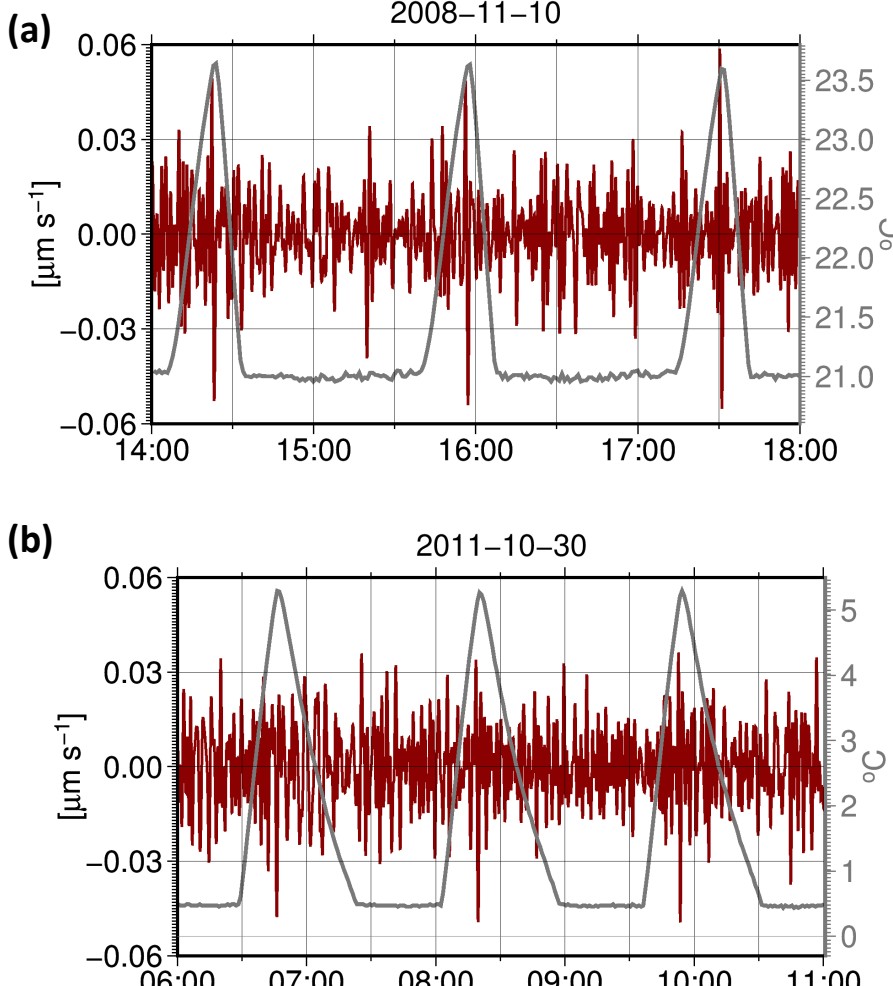

**Figure 12.** Medium timescale details of the residuals compare to the GRACE-B K-band antenna horn temperature for (a) November 2008 (during active thermal control) and (b) October 2011 (with switched-off thermal control).





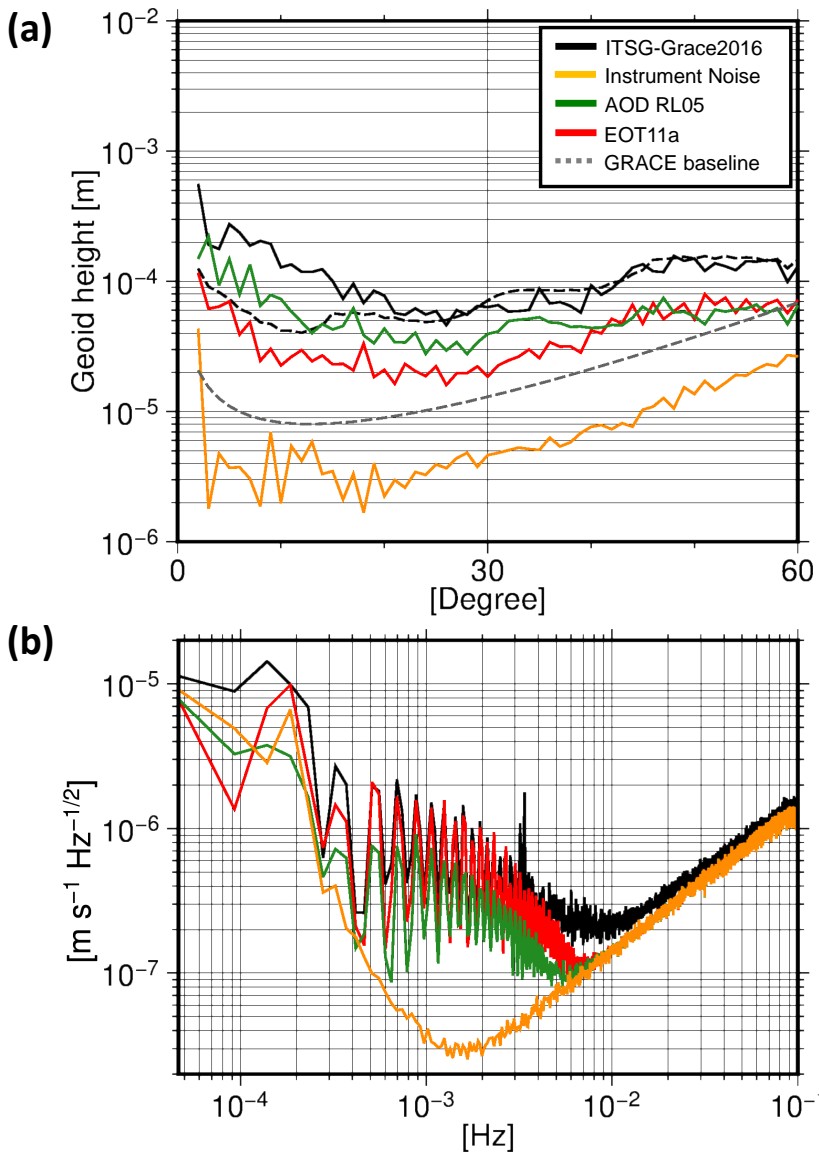

**Figure 13.** (a) RMS geoid heights per degree from simulated and real solutions with respect to the reference field GOCO05s. (b) PSD of the residuals from simulated and real data for February 2009.



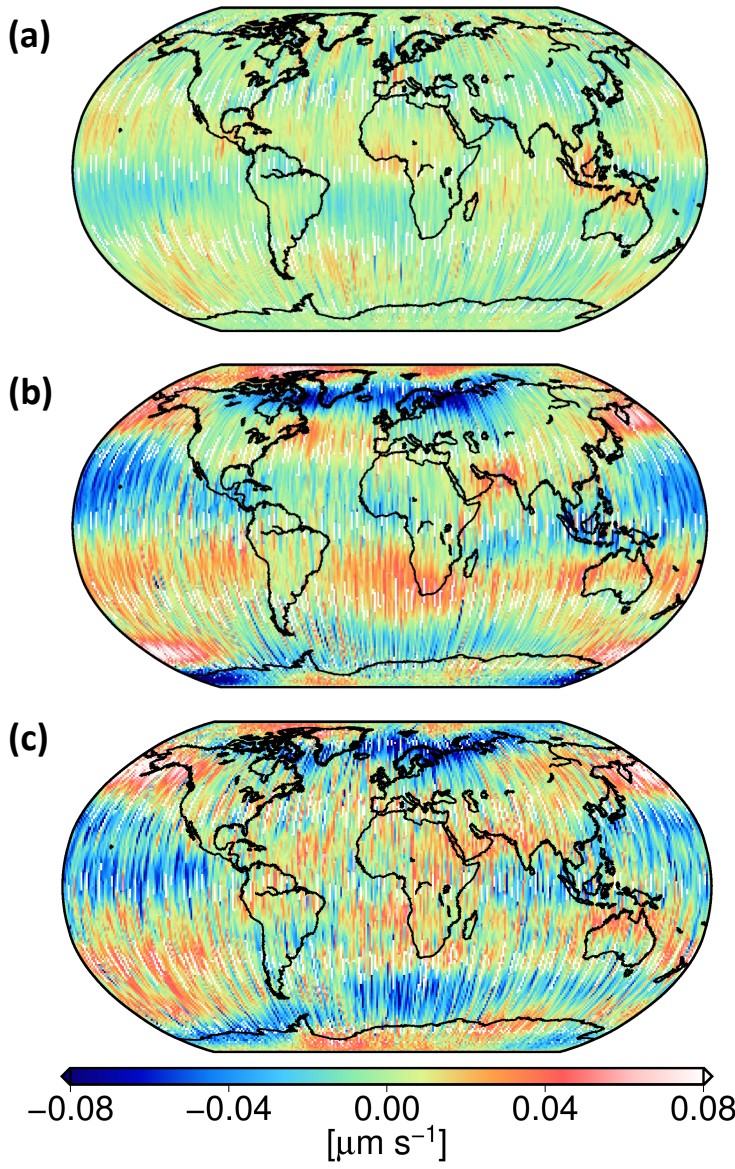

**Figure 14.** Spatial analysis of long timescale propagated errors from (a) AOD1B RL05 model and (b) EOT11a model, compared to (c) long timescale details of real residuals. The values are plotted with respect to the GRACE-A ground-track for February 2009.



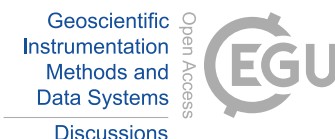

**Table 1.** Summary of ITSG-Grace2016 force models

| Perturbation | Force model | Reference |
| --- | --- | --- |
| Earth's static gravity field, trend, and annual oscillation | GOCO05S | Mayer-Gürr et al. (2015) |
| Astronomical tides (moon, sun, planets) | JPL DE421 | Folkner et al. (2009) |
| Ocean tides | EOT2011a | Savcenko and Bosch (2012) |
| Nontidal atmosphere and ocean | AOD1B RL05 | Dobslaw et al. (2013) |
| Atmospheric tides (S1, S2) | van Dam, Ray | van Dam and Ray (2010) |
| Solid earth tides | IERS2010 | Petit and Luzum (2010) |
| Pole tides | IERS2010 | |
| Ocean pole tides | IERS2010 | |
| Relativistic corrections | IERS2010 | |