# Peer review of "Multiresolution wavelet analysis applied to GRACE range rate residuals"

_Geoscientific Instrumentation, Methods and Data Systems, 2018_

## Referee Comment (RC1) · Anonymous Referee #1 · 29 Jan 2019

The twin-satellite mission GRACE has been orbiting the Earth for more than 15 years between 2002 – 2017. GRACE represents an entirely novel observing concept where a suite of very different and highly precise sensors are essential for deriving accurate time-series of time-variable global gravity fields. Many of those sensors were specifically designed for that ground-breaking mission, and even after spending 15 years on the analysis of this data record, there are still many systematics to be identified.

The single sensor most critical for reaching the objectives of the GRACE mission is the precise K-band ranging instrument which is at the focus of the present manuscript. The authors introduce an analysis frame-work based on multiresolution wavelet analysis that allows them to identify systematics in the range-rate residuals that were previously unknown to the scientific community. The paper also indicates that knowledge of those

systematics have been used successfully to tailor parametrizations used in the gravity field retrieval which contributed to a substantially reduced noise floor in the GRACE series processed by the group at TU Graz.

I therefore believe that the paper is a valuable description of a visual analytics framework for long data records of a highly sophisticated space-based instrument aiming at the observation of global change. The contribution fits well into the scope of GI. A few comments given below might nevertheless be considered in order to further shape its character as a methodological paper and make it more accessible to a larger audience. I therefore recommend that the paper should undergo a minor revision before publication.

(1) The paper introduces six examples of spatio-temporal structures that reveal either previously known or newly discovered systematics in the GRACE KBRR residuals. The view-point taken to look at the data is very different for each example, and I would recommend to find section head-lines focussing on the view-point instead of the feature identified. From my point of view, there is no need to distiguish between systematics previously known (Fig. 9 and 10) and newly discovered (all other examples) by means of specific sub-sections.

(2) I see some overlap with methods from the field of visual analytics which might be worth mentioning in the introduction. An example with some remote connection to this study has been published by Dransch et al. (2010).

(3) The summary states that the analysis methods presented here contributed in the end to the improved noise-level of ITSG2016. This claim might be substantinated by citing Chen et al.(2018), who independently validated a range different GRACE releases including ITSG2016 and found particularly low noise levels in the solutions from TU Graz.

(4) It might be worth to mention in the paper that also other sensors aboard GRACE are required to process gravity fields: Would it be benefitial to use this framework also for

accelerometer or star camera analysis? Are there any direct synergies for the analysis of other space missions as, e.g., GOCE?

A few minor points might also be considered during the revision:

(5) I'd rather prefer to use 'range-rate' instead of 'range rate'.

(6) It should be mentioned at some point that all KBRR data actually refer to the mid-point of the line-of-sight vector between GRACE-A and GRACE-B, which might be 100 km off the position of GRACE-A. For all plots shown in the paper, however, this offset can be safely neglected.

(7) p.3 l.21: There is no need to mention the degree 90 or 120 solutions, since those are not considered any further in the paper.

(8) p.5 l.15: Typo: As described...

(9) p.6 l.24: Wording suggestion: ...to prove whether or not our...

(10) p. 10 l.1: Wording suggestion: The proposed analysis framework confirms known and reveals previously unknown systematics in the residuals that allow for a specifically tailored parametrization in the gravity field retrieval.

Dransch, D., Köthur, P., Schulte, S., Klemann, V., Dobslaw, H. (2010): Assessing the quality of geoscientific simulation models with visual analytics methods - a design study. - International Journal of Geographical Information Science, 24, 10, pp. 1459-1479. DOI: http://doi.org/10.1080/13658816.2010.510800.

Chen, Q., Poropat, L., Zhang, L., Dobslaw, H., Weigelt, M., van Dam, T. (2018): Validation of the EGSIEM GRACE Gravity Fields Using GNSS Coordinate Timeseries and In-Situ Ocean Bottom Pressure Records. - Remote Sensing, 10, 12, 1976. DOI: http://doi.org/10.3390/rs10121976.

---

## Referee Comment (RC2) · Anonymous Referee #2 · 27 May 2019

GENERAL COMMENTS

The paper deals with an error analysis of the gravity field model recovery from the GRACE mission data. The MRA approach based on the discrete wavelet transform is applied to the residuals of the inversion; this method allows one to separate the residual time series into bands each having a specified frequency content. It is then possible to look separately for error sources in each of these bands. The authors first validate the method by identifying the already known error sources in short timescale details of the residuals (KBR instrument) and in the medium ones (attitude control). Based on simulations and comparing their results to real-world data, the authors then show that a possible source for long timescale details may be in an imperfect ocean tide modelling. I think this is an interesting and useful paper. The manuscript is written

in a clear way and contains new results. I recommend the publication after considering my minor comments below.

SPECIFIC COMMENTS

Page-line:

3-16: Please add the time period treated (month, year), here or later at page 6, line 22.

7-18: In Figure 11b, only a part of the time series is shown, not the complete GRACE timespan. Please modify.

8-13: Please specify over what time period the simulation was performed.

9-17: the same magnitude and spatial pattern => comparable magnitude and spatial pattern

TECHNICAL CORRECTIONS

Page-line:

2-15: contributes => contributors

2-29: no reference to Fig. 1a

5-15: descriped => described

7-14: umbra shadow => umbra

10-3: were also identified => were identified

10-19: mainly are => are mainly

---

## Author Comment (AC1) · 3 Jul 2019

We thank the reviewer for their constructive comments. We address the reviewer's comments point by point in this letter, and corresponding changes will be made to improve the manuscript.

(1) The paper introduces six examples of spatio-temporal structures that reveal either previously known or newly discovered systematics in the GRACE KBRR residuals. The view-point taken to look at the data is very different for each example, and I would recommend to find section head-lines focusing on the view-point instead of the feature identified. From my point of view, there is no need to distinguish between systematics previously known (Fig. 9 and 10) and newly discovered (all other examples) by means

of specific sub-sections.

Reply: We understand the reviewer's standpoint here and we do agree on the importance of each viewpoint in introducing our method. However, based on the comments received from the previous submission of the paper, our aim was to emphasize on new findings and insights on the potential error contributors which were not easy to conclude from classic methods.

(2) I see some overlap with methods from the field of visual analytics which might be worth mentioning in the introduction. An example with some remote connection to this study has been published by Dransch et al. (2010).

Reply: We updated the introduction section by adding "The drawback of this framework draw our attention to spatio-temporal approaches, which incorporates data analysis as well as geophysical model validation (e.g. Dransch et al. (2010))."

(3) The summary states that the analysis methods presented here contributed in the end to the improved noise-level of ITSG2016. This claim might be substantiated by citing Chen et al.(2018), who independently validated a range different GRACE releases including ITSG2016 and found particularly low noise levels in the solutions from TU Graz.

Reply: The result of the presented method contributes to the latest GRACE-only gravity field time series from TU Graz, ITSG-Grace2018.

(4) It might be worth to mention in the paper that also other sensors aboard GRACE are required to process gravity fields: Would it be beneficial to use this framework also for accelerometer or star camera analysis? Are there any direct synergies for the analysis of other space missions as, e.g., GOCE?

Reply: We thank the reviewer for highlighting these points. In the discussion section, we added this paragraph in response: "Beside the range-rate observations, the presented framework is also beneficial for the data processing of the other sensors
aboard GRACE or similar satellite missions. The results can potentially detect inconsistent time periods in each set of measurements and provide an initial interpretation of their possible origin."

(5) I'd rather prefer to use 'range-rate' instead of 'range rate'.

Reply: We updated the manuscript accordingly.

(6) It should be mentioned at some point that all KBRR data actually refer to the midpoint of the line-of-sight vector between GRACE-A and GRACE-B, which might be 100 km off the position of GRACE-A. For all plots shown in the paper, however, this offset can be safely neglected.

Reply: The systematic errors in KBR measurements are the sum of different effects from GRACE-A and/or GRACE-B and are mainly caused by the instruments onboard each satellite. Therefore, the errors can sometimes be associated with the GRACE-A or GRACE-B position or a point on the LOS vector, which is not always the mid-point. However, as mentioned by the reviewer, in our analysis this offset is negligible.

(7) p.3 l.21: There is no need to mention the degree 90 or 120 solutions, since those are not considered any further in the paper.

Reply: We applied this correction.

(8) p.5 l.15: Typo: As described. . . (9) p.6 l.24: Wording suggestion: ...to prove whether or not our. . . (10) p. 10 l.1: Wording suggestion: The proposed analysis framework confirms known and reveals previously unknown systematics in the residuals that allow for a specifically tailored parametrization in the gravity field retrieval.

Reply to 8-10: We updated the manuscript accordingly.

---

## Author Comment (AC2) · 3 Jul 2019

We thank the reviewer for their constructive comments. We address the reviewer's comments point by point in this letter, and corresponding changes will be made to improve the manuscript.

(1) 3-16: Please add the time period treated (month, year), here or later at page 6, line 22.

Reply: We have provided additional information on the treated time-span by adding "The analysis is carried out on the whole ITSG-Grace2016 time-span (April 2002 – June 2017). However, due to low data quality before 2004 and several data gaps and degraded quality of the measurements after 2016, these time periods are excluded

from the illustrations."

(2) 7-18: In Figure 11b, only a part of the time series is shown, not the complete GRACE timespan. Please modify.

Reply: We updated the manuscript accordingly.

(3) 8-13: Please specify over what time period the simulation was performed.

Reply: We followed the propositions of the reviewer and clarified this issue by adding "The simulation is carried out for the time period 2008-2009, when GRACE delivered high-quality measurements and comparison of the actual data with the output of the simulation is more relevant".

(4) 9-17: the same magnitude and spatial pattern => comparable magnitude and spatial pattern

(5) 2-15: contributes => contributors

(6) 2-29: no reference to Fig. 1a

(7) 5-15: descriped => described

(8) 7-14: umbra shadow => umbra

(9) 10-3: were also identified => were identified

(10) 10-19: mainly are => are mainly

Reply to 4-10: We updated the manuscript accordingly.